# Isaac Gym: High Performance GPU Based Physics Simulation For Robot Learning

**Viktor Makoviychuk, Lukasz Wawrzyniak, Yunrong Guo, Michelle Lu, Kier Storey,**

**Miles Macklin, David Hoeller, Nikita Rudin, Arthur Allshire, Ankur Handa, Gavriel State**

NVIDIA
{vmakoviychuk, lwawrzyniak, kellyg, michellel, kstorey, mmacklin,
dhoeller, nrudin, aallshire, ahanda, gstate}@nvidia.com

## Abstract

Isaac Gym offers a high performance learning platform to train policies for a wide variety of robotics tasks entirely on GPU. Both physics simulation and neural network policy training reside on GPU and communicate by directly passing data from physics buffers to PyTorch tensors without ever going through CPU bottlenecks. This leads to blazing fast training times for complex robotics tasks on a single GPU with 2-3 orders of magnitude improvements compared to conventional RL training that uses a CPU based simulator and GPUs for neural networks. We host the results and videos at https://sites.google.com/view/isaacgym-nvidia and Isaac Gym can be downloaded at https://developer.nvidia.com/isaac-gym. The benchmark and environments are available at https://github.com/NVIDIA-Omniverse/IsaacGymEnvs.

## 1 Contributions

- Development of high-fidelity GPU-accelerated robotics simulator for robot learning tasks. With tools to load commonly used robot description formats - URDF and MJCF. A Tensor API in Python providing direct access to physics buffers by wrapping them into PyTorch tensors without going through any CPU bottlenecks.

- We achieve significant speed-ups in training various simulated environments: Ant and Humanoid environments can achieve performant locomotion in 20 seconds and 4 minutes respectively, ANYmal [9] in under 2 minutes, Humanoid character animation using AMP [23] in 6 minutes and cube rotation with Shadow Hand in 35 minutes all on a **single NVIDIA A100 GPU**. Additionally, we reproduce OpenAI Shadow Hand cube training setup and show that we can achieve 20 consecutive successes with feed forward and 40 consecutive successes with LSTM networks with a success tolerance of 0.4 rad in about 50 minutes and 3 hours on average respectively on A100. In contrast, OpenAI effort required 30 hours and 17 hours respectively on a combination of a CPU cluster (384 CPUs with 16 cores each) and 8 NVIDIA V100 GPUs with MuJoCo [30] using a conventional RL training setup.

- Recent successful sim-to-real transfer results on ANYmal and TriFinger further showcase the ability of our simulator to perform high-fidelity contact rich manipulation.

## 2 Introduction

In recent years, reinforcement learning (RL) has become one of the most promising research areas in machine learning and has demonstrated great potential for solving sophisticated decision-making

35th Conference on Neural Information Processing Systems (NeurIPS 2021) Track on Datasets and Benchmarks.

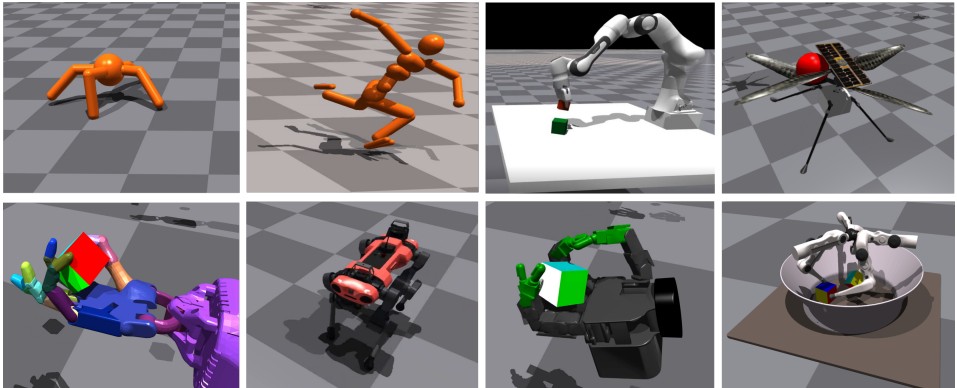

Figure 1: Isaac Gym allows high performance training on a variety of robotics environments. We benchmark on 8 different environments that offer a wide range of complexity and show the strengths of the simulator in blazing fast policy training on a single GPU. *Top*: Ant, Humanoid, Franka-cube-stack, Ingenuity. *Bottom*: ShadowHand, ANYmal, Allegro, TriFinger.

problems. Simulators play a key role in training robots improving both the safety and iteration speed in the learning process. To date, most researchers have relied on a combination of CPUs and GPUs to run reinforcement learning system [21]. Different parts of the computer tackle different steps of the physics simulation and rendering process. CPUs are used to simulate environment physics, calculate rewards, and run the environment, while GPUs are used to accelerate neural network models during training and inference as well as rendering if required.

Popular physics engines like MuJoCo[30], PyBullet[6], DART[12], Drake[28], V-Rep[25] *etc.* need large CPU clusters to solve challenging control tasks naturally. For instance, in [2], almost 30,000 CPU cores (920 worker machines with 32 cores each) were used to train a robot to solve the Rubik's Cube task using RL. One way to speed-up simulation and training is to make use of hardware accelerators. GPUs have enjoyed enormous success in computer graphics are also naturally suited for highly parallel simulations. This approach was taken by [13], and showed very promising results running simulation on GPU, proving that it is possible to greatly reduce both training time as well as computational resources required to solve very challenging tasks using RL. However, some bottlenecks were still not addressed in the work – simulation was on GPU but physics state was copied back to CPU. There, observations and rewards were calculated using optimized C++ code and later copied back to GPU where policy and value networks ran.

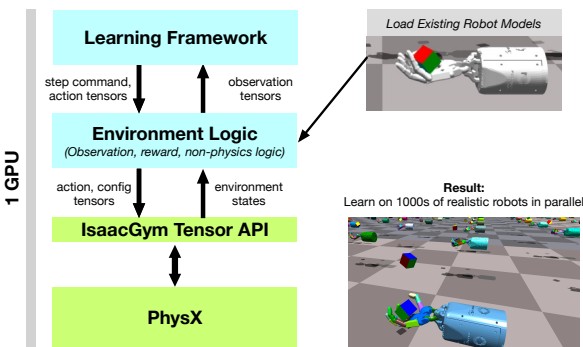

Figure 2: An illustration of the Isaac Gym pipeline. The Tensor API provides an interface to Python code to step the PhysX backend, as well as get and set simulator states, directly on the GPU, allowing a 100-1000x speedup in the overall RL training pipeline while providing high-fidelity simulation and the ability to interface with existing robot models.

To address these bottlenecks, we present **Isaac Gym** - an end-to-end high performance robotics simulation platform. It runs an end-to-end GPU accelerated training pipeline, which allows researchers to overcome the aforementioned limitations and achieves 100x-1000x training speed-up in continuous control tasks. Isaac Gym leverages NVIDIA PhysX [18] to provide a GPU-accelerated simulation back-end, allowing it to gather experience data required for robotics RL at rates only achievable using a high degree of parallelism. It supports a Py-Torch tensor-based API to access the results of physics simulation natively on the GPU. Observation tensors can be used as inputs to a policy network and the resulting action tensors can be directly fed back into the physics sys-

tem. We note that others [8] have recently begun attempting an approach similar to Isaac Gym with respect to running end-to-end training on hardware accelerators. Isaac Gym provides a straightforward API for creating and populating a scene with robots and objects, supporting loading data from the common URDF and MJCF file formats. Each environment is duplicated as many times as needed, while preserving the ability for variations between copies (*e.g.* via Domain Randomization [29]). Environments are simulated simultaneously in parallel without interaction with other environments. Using a fully GPU-accelerated simulation and training pipeline can help lower the barrier for research, enabling solving of tasks with a single GPU that were previously only possible on massive CPU clusters. We provide training examples with highly optimized Proximal Policy Optimization (PPO). While the included examples use PyTorch, users should also be able to integrate with TensorFlow training libraries with further customization. An overview of the system is provided in Figure 2.

## 3   Physics Simulation

Robots are simulated using PhysX [18] reduced coordinate articulations. Any individual rigid bodies may be simulated using either maximal coordinate rigid bodies or single-link reduced coordinate articulations. Articulations with a single link and rigid bodies are equivalent and interchangeable.We use the Temporal Gauss Seidel (TGS)  [14] solver to compute the future states of objects in our physics simulation. More detailed description can be found in: A.1

## 4   Environments

We implemented a diverse set of environments covering different application areas. Here we describe a subset of representative examples and key points related to the training.  Benchmark results on the simulation performance and training results are presented in the subsequent sections.  All environments are trained using the Proximal Policy Optimization algorithm [27], using rl_games, a highly-optimized GPU end-to-end implementation from [15].  This implementation vectorizes observations and actions on GPU allowing us to take advantage of the parallelization provided by the simulator. We list the environments used in our experiments below:

1. **Locomotion Environments:** Ant, Humanoid, Ingenuity, ANYmal
2. **Franka Cube Stacking**
3. **Humanoid Character Animation**
4. **Robotic Hands:** Shadow, Allegro, Trifinger

Unless stated otherwise, all experiments are done on a **single A100 GPU**. All training runs for each environment are **averaged over 5 seeds**. The reward curves are plotted with $\mu \pm \sigma$ regions. All the environments by default follow symmetric actor-critic approach with shared observations as well as shared network for policy and value functions. Sharing the network allows faster forward passes and improves training. Moreover, for Shadow Hand and TriFinger, we also use an asymmetric actor critic approach [24] with policy observations that are closest to real world settings while value function receives privileged state information from simulation as well as the observations received by the policy. This approach is naturally suited for sim-to-real transfers. Detailed hyper-parameters for each training task are in Table 14. Rewards and observations for each environment can be found in A.2.

## 5   Characterising Simulation Performance

We first characterise the simulation performance as a function of number of environments. As we vary this number, we aim to keep the overall experience an RL agent observes constant by decreasing the horizon length proportionally (*i.e.* number of steps in PPO) for a fair comparison. While we provide detailed training studies for many environments later, we characterise simulation performance only for **Ant**, **Humanoid** and **Shadow Hand** as they are sufficiently complex to test the limits of the simulation and also represent a gradual increase in the complexity. All three environments use feed forward networks for training.

### 5.1   Ant

| Environment | Control Type | Sim $dt$ | Control $dt$ | Action Dims |
|---|---|---|---|---|
| Ant | Joint Torques | 1/120 s | 1/60 s | 8 |
| Humanoid | Joint Torques | 1/120 s | 1/60 s | 21 |
| Ingenuity | Rigid Body Forces | 1/200 s | 1/200 s | 6 |
| ANYmal | Joint Position Targets | 1/200 s | 1/50 s | 12 |
| Franka Cube Stacking | Operation Space Control | 1/60 s | 1/60 s | 7 |
| Shadow Hand Standard | Joint Position Targets | 1/120 s | 1/60 s | 20 |
| Shadow Hand OpenAI | Joint Position Targets | 1/120 s | 1/20 s | 20 |
| Allegro Hand | Joint Position Targets | 1/120 s | 1/20 s | 16 |
| TriFinger | Joint Torques | 1/200 s | 1/50 s | 9 |

Table 1: Simulation setup for the environments.

We first experiment with the standard Ant environment where the agent is trained to run on a flat ground. We find that as the number of agents is increased, the training time, as expected, is reduced *i.e.* changing the number of environments from 256 to 8192 — an increase by 5 orders of magnitude — leads to a reduction in training time to reach 7000 reward by an order of magnitude from 1000 seconds (~16.6 minutes) to 100 seconds (~1.6 minutes). **However, note that Ant reaches performant locomotion at 3000 reward in just 20 seconds on a single GPU.**

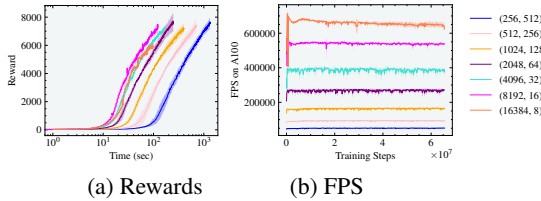

(a) Rewards      (b) FPS

Figure 3: Rewards and effective FPS for the Ant environment with respect to number of parallel environments. Best training time is achieved with 8192 environments and a horizon lengths of 16.

Since Ant is one of the simplest environments to simulate, the number of parallel environment steps per second as depicted in the Figure 3(b) can go as high as 700K. We do not observe gains when increasing the number of environments from 8192 to 16384 due to reduced horizon length.

## 5.2 Humanoid

The Humanoid environment has more degrees of freedom and requires the agent to discover the gait that lets itself balance on two feet and walk on the ground. As observed in Figure 4, the training times are increased by an order of magnitude compared to the Ant in Figure 3.

We also note in Figure 4 that as the number of agents is increased, in this case, from 256 to 4096, the training time needed to reach the highest reward of 7000 is reduced by an order of magnitude from $10^4$ seconds (~2.7 hours) to $10^3$ seconds (~17 minutes). **However, performant locomotion starts happening at around a reward of 5000 at a training time of just 4 minutes.** Going beyond 4096

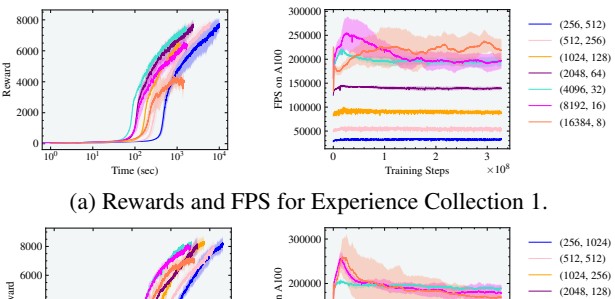

(a) Rewards and FPS for Experience Collection 1.

(b) Rewards and FPS for Experience Collection 2.

Figure 4: Rewards and FPS with respect to number of parallel environments for the Humanoid. Best training time is achieved with 4096 environments and a horizon lengths of 32.

environments for this set up resulted in no further gains and in fact led to both increase in training time and sub-optimal gaits. We attribute this to the complexity of the environment that makes it challenging to learn walking at such small horizon lengths.

We verified this by training on another set of environment and horizon length combinations where horizon length was increased by a factor of 2 compared to Figure 4(a). As shown in the Figure 4(b), the humanoid is able to walk even with 8192 and 16384 environments which have small horizon

lengths of 32 and 16 respectively but sufficiently long to enable learning. Also worth noting that due to the increased degrees of freedom the number of parallel environment steps per second is reduced from 700K for Ant to 200K for Humanoid as shown in Figure 4.

### 5.3 Shadow Hand

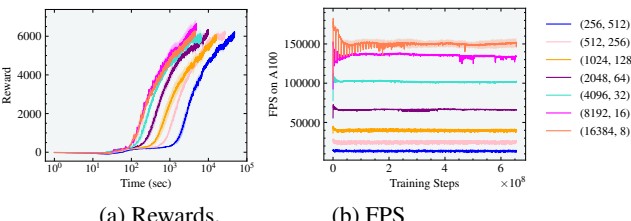

|  | (256, 512) |
|--|--|
|  | (512, 256) |
|  | (1024, 128) |
|  | (2048, 64) |
|  | (4096, 32) |
|  | (8192, 16) |
|  | (16384, 8) |

(a) Rewards.          (b) FPS

Figure 5: Rewards and effective FPS with respect to number of parallel environments for the shadow hand environment. Best training time is achieved with both 8192 and 16384 environments and horizon lengths of 16 and 8 respectively.

Lastly, we experiment with Shadow Hand [21] to learn to rotate a cube resting on the palm to a target orientation using the fingers and the wrist. This task is challenging due to the number of DoFs involved and the contacts that are made and broken during the process of rotation. Our results with Shadow Hand environment follow similar trends. As the number of agents is increased, in this case, from 256 to 16384, the training time is reduced by an order of magnitude from $5 \times 10^4$ seconds (~14 hours) to $3 \times 10^3$ seconds (~1 hour). **We find that the environment reaches performant dexterity of 10 consecutive successes at reward of 3000 in just 5 minutes.**[1] Further performance improvements continue to happen as more experience is collected. Additionally, we find that the horizon length of 8 for 16384 agents still allows learning re-posing the cube. The maximum effective frame-rate of 150K number of parallel environment steps per second was achieved with 16384 agents.

## 6  Experiments with RL Training

We provide details and performance metrics for environments mentioned in Section 4 trained using a PPO implementation that operates on vectorised environments.

### 6.1  Locomotion environments

**Ant**  The Ant [19] model has four legs with two degrees of freedom per leg. On A100 with 4096 agents simulated in parallel we find that ant can learn to run and achieve a reward above 3000 in just 20 seconds, and fully converge in under 2 minutes. The average simulation performance achieved during training is 540K environment steps per second. The results are shown in Figure 6(a). For details of the reward function and the observations used, we refer to Appendix A.2.1.

**Humanoid**  The Humanoid environment [7] has 21 DOFs and on a A100 with 4096 agents simulated in parallel we can train it to run — a reward threshold of 5000 — in less than 4 minutes. This is 4x faster than our previous results in [13] obtained using the same threshold. As shown in Figures 4, we achieve peak performance for this environment at 4096 agents. Figure 6(b) shows the evolution of reward as a function of time. The reward function and the observations used are described in Appendix A.2.1.

**Ingenuity**  We train a simplified model of NASA's Ingenuity helicopter [17] to navigate to a target that periodically teleports to different locations. The environment with trained with 4096 agents and achieves a reward of 5000 in just under 30 seconds. Forces are applied directly to the two rotors on the chassis, rather than simulating aerodynamics. We use a martian gravity of -3.721 $m/s^2$.

**ANYmal Robot Locomotion**  ANYmal [5] is a four-legged dog-like robot, and has been used for experiments on navigation of rough and variable terrain. The task is to follow target X, Y, and yaw base velocities while minimizing joint torques. The target velocities are randomized at each reset and are provided as observations alongside the positional and angular velocities of the base, the measured gravity vector, most recent actions, and DOF positions and velocities. With 4096 agents simulating in parallel, we can train robot in under 2 minutes. The reward function is defined in A.2.2

---

[1]The experiments used Shadow Hand Standard variant as explained in Section 6.4.1.

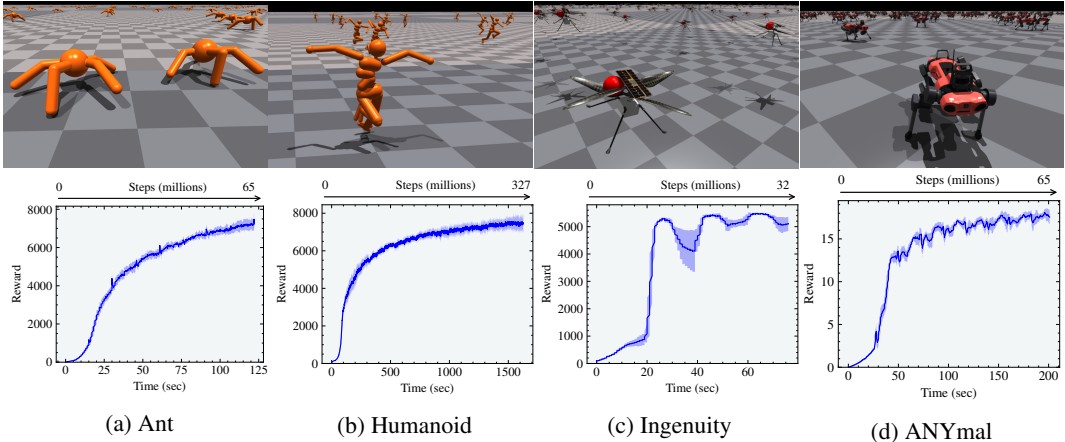

|(a) Ant|(b) Humanoid|(c) Ingenuity|(d) ANYmal|

Figure 6: Locomotion environments and the corresponding reward curves.

**ANYmal Sim-to-real on Uneven Terrain** In addition to the simple flat terrain environment, we have developed a rough terrain locomotion task for ANYmal and transferring trained policies to the real robot. The robot learns to walk on uneven surfaces, slopes, stairs and obstacles.

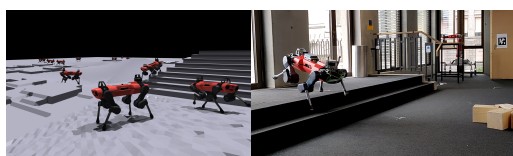

Figure 7: Trained policy for ANYmal on rough terrain tested in simulation and on the real robot.

In addition to the observations of the flat terrain, it receives terrain height measurements around the robot's base. For sim-to-real transfer we extend the reward function, add noise to the observations, randomize the friction coefficient of the ground, randomly push the robots during the episode and add an actuator network to the simulation. Following the approach used in [10], the actuator network is trained to model the complex dynamics of the series elastic actuators of the real robot. We used curriculum - the robots start to learn on simple versions of the terrains, and when they are able to solve a certain level the difficulty is automatically increased. With 4096 environments, we can train the full task and transfer to the real robot in under 20 minutes.

## 6.2 Humanoid Character Animation

We evaluate the performance of Isaac Gym on adversarial imitation learning tasks using an implementation of adversarial motion priors (AMP) [23]. This technique enables physically simulated humanoid character to imitate complex behaviors from reference motion data. Instead of a manually engineered imitation objective, as is commonly used in prior systems [22],

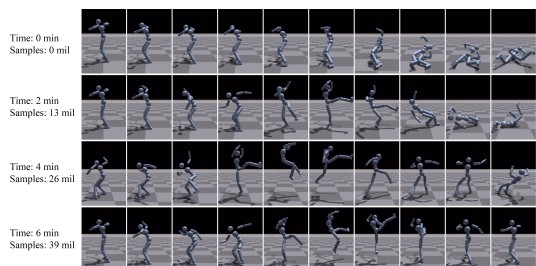

Figure 8: Humanoid character trained using AMP to imitate a spin-kick.

AMP learns an imitation objective using an adversarial discriminator trained to differentiate between motion from the dataset and motions produced by the policy. Our character is modelled as a 34-DOF humanoid, and all motion clips are recorded from human actors using motion capture. Table 8 in Section A.2.2 details the observation features.

The adversarial training process enables the character to closely imitate a diverse corpus of motions, ranging from common locomotion behaviors, such as walking and running, to more athletic behaviors, such as spin-kicks and dancing. Effective policies can be learned with approximately 39 million samples, requiring approximately **6 minutes** with 4096 environments. The implementation provided by Peng *et al.*, 2021 [23] requires about **1 day** (30 hours) on on 16 CPU

cores to simulate a similar number of samples in PyBullet. Therefore, Isaac Gym provides **300x or 2.48 orders of magnitude improvement** in the training time.

### 6.3 Franka Cube Stacking

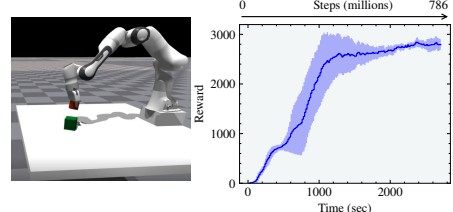

Figure 9: The Franka Cube Stacking environment and the corresponding reward curves.

We use 16384 agents to train a Franka robot to stack a cube on top of an other. In this environment, we use a slightly different choice of action space, Operation Space Control (OSC), for learning. OSC [11] is a task-space compliant controller that has been shown to enable faster policy learning compared to joint-space controllers [33] and learn contact-rich tasks [16]. We obtain convergence with this controller in under 25 minutes. Figure 9 shows the training results. More details about using (differentiable) OSC control for solving challenging robotics tasks can be found in [31]

### 6.4 Robotic Hands

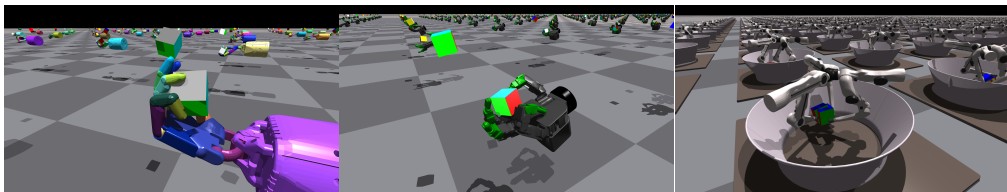

Figure 10: Three in-hand manipulators implemented in Isaac Gym: Shadow Hand, Trifinger, and Allegro

Large-scale simulation has the ability to solve not just individual instances but whole classes of problems in robotics, by leveraging the generality of the model-free reinforcement learning framework. Dexterous manipulations is one of the most challenging problems in robotics. To show the performance of our simulator and the ability to realistically model contact we implemented 3 different hand training environments. Firstly, the Shadow Dexterous Hand[20]. We follow the standard formulation where policy and value function both receive the same input as well as OpenAI observations with asymmetric formulation and domain randomisation from [21]. Secondly, the TriFinger robot [32], which shows the ability to do 6-DoF manipulation by reposing the cube to a desired position and orientation, a task which has previously shown to be challenging for model-free reinforcement learning [4]. We use asymmetric actor-critic and domain randomisation for TriFinger and demonstrate sim-to-real transfer on a real robot. Finally, we reuse system from the Shadow Hand to the Allegro hand [26] with minimal changes to show the generality of our approach. These three environments are depicted in Figure 10 and the corresponding reward curves in Figure 11.

#### 6.4.1 Shadow Hand

As mentioned, the task with Shadow Hand is to manipulate the cube to achieve a specific target orientation and is inspired by OpenAI *et al.* [21]. We train with multiple variants on the Shadow Hand environment and describe them below:

**Shadow Hand Standard** In this setting, we use a standard formulation for training where the policy and the value function use feed forward networks and receive the same input observations. The default observations we used for the Shadow Hand Standard include joint position, velocities, forces, force-torque sensors reading from each fingertip, manipulated object position and orientation, linear and angular velocities, goal orientation, relative rotation between the current object and target rotations, actions applied on the previous step. For a detailed overview of observation and reward, see Appendix A.4. Also note that this variant does not use any randomizations.

**Shadow Hand OpenAI** We also reproduce results with OpenAI Shadow Hand experiments in Isaac Gym with observations used in dexterity work from OpenAI *et al.* [21]. A key difference

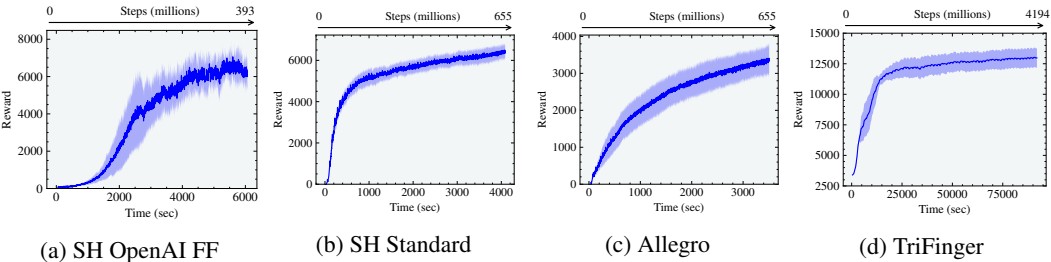

| (a) SH OpenAI FF | (b) SH Standard | (c) Allegro | (d) TriFinger |

Figure 11: Reward curves for the three in-hand manipulation environments implemented in Isaac Gym. These results are obtained with **(a)** Showdow Hand OpenAI FF **(b)** Shadow Hand Standard **(c)** Allegro and **(d)** TriFinger. Shadow Hand OpenAI and TriFinger are trained with asymmetric actor-critic and domain radomisation while Shadow Hand Standard and Allegro are trained with standard observations and symmetric actor-critic with no domain randomisation.

between this and the Shadow Hand Standard variant is that it uses asymmetric observations. The policy receives only the input observations that are possible to obtain in the real world settings while the value function receives the same observations in addition to the other privileged information available from the simulator. This variant should make it possible to transfer the policy to the real world, mimicking the setup in [21]. The observations for the policy and value function are provided in Table 11. We experiment with both feed forward networks (SH OpenAI FF) and LSTMs (SH OpenAI LSTM). The LSTM networks are trained with a sequence length of 4. It is worth noting that only networks trained with OpenAI observations use domain randomisation to closely match the results in OpenAI dexterity work [21].

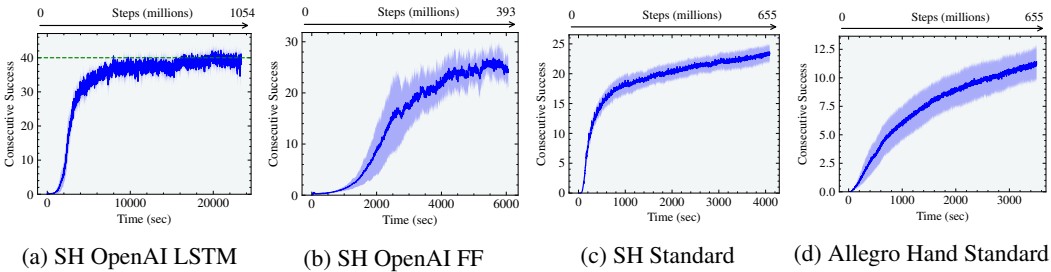

| (a) SH OpenAI LSTM | (b) SH OpenAI FF | (c) SH Standard | (d) Allegro Hand Standard |

Figure 12: Consecutive successes per episode for **(a)** Shadow Hand with OpenAI observation and LSTMs, **(b)** Shadow Hand with OpenAI observation and feed forward networks **(c)** Shadow Hand with Standard observations and **(d)** Allegro Hand with Standard observations. Shadow Hand Standard and Allegro Hand Standard both use feed forward networks for policy and value functions.

**Randomizations** For domain randomization we closely followed the approach proposed in [21] and applied correlated and uncorrelated noise to observations, actions, as well as randomized cube size and all the key physics properties – masses, inertia tensors, friction, restitution, joint limits, stiffness and damping. Full details of these are available in Appendix A.4.1.

Figure 11(a), (b) and (c) show the reward curves for various settings we used for Shadow Hand. Shadow Hand Standard — trained with no randomization and uses symmetric actor critic setting with a feed forward network — is the fastest to reach a reward of 6000. This setting achieves 20 consecutive successes in under 35 minutes. Important to remember that this setting is not suitable for sim-to-real transfer as it includes some observations that may not be directly available in the real world. We now focus on experiments with OpenAI observations and asymmetric feed-forward actor-critic. This setting is suited for sim-to-real transfer and the policy uses only the observations that are possible to obtain in the real world. As shown in Figure 12(b), we achieved more than 20 consecutive successes in less than 1 hour. In contrast, for the same performance it takes 30 hours on the OpenAI setup consisting of CPU based simulation and training setup running MuJoCo [30] simulator on a cluster of 384 16-core CPUs with 6144 CPU cores in total and using 8 NVIDIA V100 GPUs for training. In Figure 12(a) we show that using LSTM networks, the performance increases and we can reach 40 consecutive successes within 3 hours while OpenAI *et al.* [21] achieve same

performance in ~20 hours. Since OpenAI *et al.* [21] show results only with 1 seed, comparing their result with our best seed we note that 40 consecutive successes with LSTM experiments can be achieved in just 2.5 hours. We provide the results for Shadow Hand OpenAI experiment with success tolerance of 0.1 in the Appendix A.4.

### 6.4.2 TriFinger

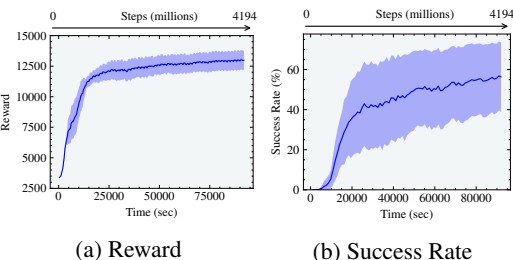

(a) Reward      (b) Success Rate

Figure 13: TriFinger reward and success rate.

The TriFinger manipulation task, originating in [32], involves picking a cube lying on a flat surface and repositioning it to a desired 6-degrees-of-freedom pose. The manipulator has 3 fingers each with three degrees of freedom. In [3], it was shown that Isaac Gym training combined with Domain Randomization allows sim-to-real transfer. The environment is shown in Figure 10. We show the reward and success rate in simulation in Figure 13. [3] transfer results from simulation to the real world and note that success rate in the real world is 55%. We refer to [3] for more detailed analysis. In particular, this example shows the ability of policies learned using Isaac Gym's physics to generalize to the real world. It is worth noting that the robot is situated in a different location and therefore the sim-to-real transfer was done remotely.

### 6.4.3 Allegro Hand

We learn cube orientation with Allegro Hand and use the same reward as for the Shadow Hand as well similar observation scheme, with the only difference — smaller number of observations because of the different number of fingers in Allegro Hand — that it has 4 fingers instead of 5 and fewer degrees of freedom as a result, shown in Appendix A.2.3. Figure 11(d) shows the reward curves for Allegro Hand and Figure 12(d) shows consecutive successes achieved. Interestingly, despite having fewer degrees of freedom this hand does not achieve as high consecutive successes as Shadow hand. This is because the wrist is fixed and fingers are slightly longer. We observed in Shadow hand experiment that having a movable wrist allows for better manipulation when reorienting the cube.

## 7 Limitations

A number of limitations and constraints exist with our current implementation. Maximum acceleration of the training process can be achieved only when simulating thousands of environments in parallel for challenging tasks. For simpler tasks or fewer environments, GPU accelerated end-to-end simulation may provide only a minor performance improvement, or none at all.

Also, in some cases, fully deterministic simulation of all environments may not be possible. While the experiments we describe above are deterministic on the same system across multiple runs, we have observed non-deterministic training when changing scale and mass at run-time in the Shadow Hand environment. Due to GPU work scheduling, some run-time changes to simulation parameters can alter the order in which operations take place, as environment updates can happen while the GPU is doing other work. Because of the nature of floating point numeric storage, any alteration of execution ordering can cause small changes in the least significant bits of output data, leading to divergent execution over the simulation of thousands of environments and simulation frames. To avoid this limitation, we randomize scale and mass at startup, but do not re-randomize these specific parameters at reset. We still have excellent coverage of the randomization range due to the fact that many thousands of environments are used. Finally, using our current tensor API it's not possible to add new actors into an already-running simulation. We expect to address many of these constraints in the future.

## 8 Social Impact

Isaac Gym enables researchers with only a local workstation to run experiments that were previously possible only with expensive, energy intensive clusters. We hope that the acceleration it enables

for RL training will lower barriers to entry for RL research, allowing for wider participation by previously underrepresented groups. Energy usage for training existing environments should also decrease dramatically. By our estimates, training the OpenAI ShadowHand task on GPU with Isaac Gym consumes about 1/300 the electricity required to train vs OpenAI's CPU version. The downside risk of efficiency improvements is that we may encourage increased use of energy for training more complex RL tasks, or through greater uptake of RL research in the research community. We also need to consider the long term impact that improved robotics may have on the automation of tasks previously only possible with human labour. We hope that these efforts will reduce the need to put humans in dangerous situations, ultimately saving lives, but there is also potential for worker displacement.

## Acknowledgments and Disclosure of Funding

We would like to thank the following for additional hard work helping us with this work.

Jonah Alben, Rika Antonova, Ayon Bakshi, Dennis Da, Shoubhik Debnath, Clemens Eppner, Animesh Garg, Renato Gasoto, Isabella Huang, Andrew Kondrich, Rev Lebaredian, Qiyang Li, Jacky Liang, Denys Makoviichuk, Brendon Matusch, Hammad Mazhar, Mayank Mittal, Adam Moravansky, Yashraj Narang, Fabio Ramos, Andrew Reidmeyer, Philipp Reist, Tony Scudiero, Mike Skolones, Balakumar Sundaralingam, Liila Torabi, Cameron Upright, Zhaoming Xie, Winnie Xu, Yuke Zhu, and the rest of the NVIDIA PhysX, Omniverse, and robotics research teams. We also thank Jason Peng and Josiah Wong for the help in AMP and Franka Cube Stacking experiments.

This work was fully funded by NVIDIA Corporation.

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
