# OpenReview forum: "Isaac Gym: High Performance GPU Based Physics Simulation For Robot Learning"
_NeurIPS.cc/2021/Track/Datasets_and_Benchmarks/Round2 — NeurIPS 2021 Datasets and Benchmarks Track (Round 2)_

### Official Review · Reviewer_5pxa · 2021-09-20
**A powerful GPU-based system for Robot Learning**

**Rating:** 7
**Confidence:** 4
**Correctness:** Good
**Clarity:** Good

**Strengths:**

This paper is well written. The authors studied a very important issue. Isaac Gym is a very impressive system for Robot learning. The design I like most is the Tensor API in Python providing direct access to physics buffers by wrapping them into PyTorch tensors without going through any CPU bottlenecks. This can significantly improve the performance and also be very easy to use. It will also support a variety of environment sensors - position, velocity, force, torque, etc.

**Weaknesses:**

However, there are also some concerns in this paper. As mentioned by the authors, maximum acceleration of the training process can be achieved only when simulating thousands of environments in parallel for challenging tasks. For simpler tasks or fewer environments, GPU accelerated end-to-end simulation may provide only a minor performance improvement, or none at all. In some cases, fully deterministic simulation of all environments may not be possible.

**Additional Feedback:**

N/A

**Documentation:**

OK

**Relation To Prior Work:**

Good

**Summary And Contributions:**

In this paper, the authors designed and implemented Isaac Gym, which is a powerful system that enables the users to train policies for a wide variety of robotics tasks entirely on GPU.

This paper is well written. The authors studied a very important issue. Isaac Gym is a very impressive system for Robot learning. The design I like most is the Tensor API in Python providing direct access to physics buffers by wrapping them into PyTorch tensors without going through any CPU bottlenecks. This can significantly improve the performance and also be very easy to use. It will also support a variety of environment sensors - position, velocity, force, torque, etc.

However, there are also some concerns in this paper. As mentioned by the authors, maximum acceleration of the training process can be achieved only when simulating thousands of environments in parallel for challenging tasks. For simpler tasks or fewer environments, GPU accelerated end-to-end simulation may provide only a minor performance improvement, or none at all. In some cases, fully deterministic simulation of all environments may not be possible.

---

> ### Author Response · Authors · 2021-09-30
> **Response to Reviewer 5pxa**
>
> We thank the reviewer for the positive review, and address the couple of specific weaknesses point by point below:
>
> **As mentioned by the authors, maximum acceleration of the training process can be achieved only when simulating thousands of environments in parallel for challenging tasks. For simpler tasks or fewer environments, GPU accelerated end-to-end simulation may provide only a minor performance improvement, or none at all.**
>
> Even in the case of simpler and/or a smaller number of environments, such as for the Ant environment, Isaac Gym still will provide noticeable speed-ups and training time reductions. With just 256 Ant environments a significant improvement over CPU based simulation can be observed. With our end-to-end GPU accelerated pipeline we also eliminated the time required for data copying from CPU to GPU and back for policy inference. For the Ant environment, our experiments show gains as the number of environments simulated is increased into the thousands (Figure 3a).
>
> **In some cases, fully deterministic simulation of all environments may not be possible.**
>
> It can be challenging to achieve fully deterministic behavior on GPUs due to the fact that floating point operations are order dependent, and due to the highly threaded nature of GPU program execution, execution order cannot always be guaranteed without the use of synchronization primitives, which can lower performance. This is true not only in the Isaac Gym physics simulation code, but also in the code used for neural network training. A more detailed discussion of the latter aspect of the problem can be found here: https://pytorch.org/docs/stable/notes/randomness.html
>
> On the physics simulation side, since our original submission, we did find one instance where multiple streams of compute work were taking place simultaneously without synchronization primitives to enforce execution order. With additional synchronization in place we have been able to observe determinacy up to approximately 13 million environment-steps with the Shadow Hand environment, but non-determinacy begins after that point. It is possible that there are other instances where additional synchronization is needed to achieve fully deterministic training.
>
> Note that regardless of whether we can achieve fully deterministic training, we have shown multiple sim2real results, which suggests lack of determinism is not harmful for training.
>
> We are continuing to investigate other areas where additional synchronization may be appropriate and will update this section for the camera ready paper once we can confirm any more significant improvements to determinacy.

---

### Official Review · Reviewer_jNvF · 2021-09-20
**Great contribution to the robotics community**

**Rating:** 8
**Confidence:** 3

**Strengths:**

1. Characterization of simulation performance in isolation (# of environments) on 3 tasks
2. Numerous benchmarks of RL performance on a variety of standard tasks.
3. Substantial speedups over the existing state-of-the-art on all benchmarks.


**Weaknesses:**

There is insufficient explanation of the cause of the non-deterministic behavior observed on the shadow hand experiment. Is it due to the domain randomization is some way?

**Additional Feedback:**

The figures in section 5 are difficult to read.


**Clarity:**

The paper is well written.


**Correctness:**

Yes the submission is correct and the experimental design is appropriate and properly performed.

**Documentation:**

Isaac Gym is currently available for preview release and the benchmarks from the paper are listed on github but seemingly not yet available.


**Ethics:**

The paper discusses the social impact of the paper, indicating that Isaac Gym has the potential to reduce the energy consumption of RL research but this may be balanced out by the increase in the complexity of robotics research. The discussion of ethics is sufficient.


**Relation To Prior Work:**

The paper provides a satisfactory discussion of the related work.


**Summary And Contributions:**

The paper presents Isaac Gym, a gpu accelerated physics simulation that is used for training policies for robotics via reinforcement learning. Isaac gym substantially decreasing the time to solution for RL tasks by running and optimizing the physics simulation for the GPU and removing any CPU bottlenecks caused by the communication between simulation and training. The paper also demonstrates that their simulator is capable of training policies that can transfer to real robots in real environments.

---

> ### Author Response · Authors · 2021-09-30
> **Reponse to Reviewer  jNvF**
>
> Thank you for your encouraging review. We address some specific points below.
>
> **There is insufficient explanation of the cause of the non-deterministic behavior observed on the shadow hand experiment. Is it due to the domain randomization in some way?**
>
> The domain randomizations are not directly to blame in this case.
>
> It can be challenging to achieve fully deterministic behavior on GPUs due to the fact that floating point operations are order dependent, and due to the highly threaded nature of GPU program execution, execution order cannot always be guaranteed without the use of synchronization primitives, which can lower performance. This is true not only in the Isaac Gym physics simulation code, but also in the code used for neural network training. A more detailed discussion of the latter aspect of the problem can be found here: https://pytorch.org/docs/stable/notes/randomness.html
>
> On the physics simulation side, since our original submission, we did find one instance where multiple streams of compute work were taking place simultaneously without synchronization primitives to enforce execution order. With additional synchronization in place we have been able to observe determinacy up to approximately 13 million environment-steps with the Shadow Hand environment, but non-determinacy begins after that point. It is possible that there are other instances where additional synchronization is needed to achieve fully deterministic training.
>
> Note that regardless of whether we can achieve fully deterministic training, we have shown multiple sim2real results, which suggests lack of determinism is not harmful for training.
>
> We are continuing to investigate other areas where additional synchronization may be appropriate and will update this section for the camera ready paper once we can confirm any more significant improvements to determinacy.
>
>
>
>
> **The benchmarks from the paper are listed on github but seemingly not yet available**
>
> Several of the benchmarks listed in the paper are available within the current Isaac Gym Preview 2 download release. These include:
>
> - Ant
> - Humanoid
> - Ingenuity
> - ShadowHand (basic)
> - ShadowHand (DR / Asymmetric OpenAI observations with Feed Forward network)
> - Anymal (flat terrain only)
>
> We are in the process of finalizing an updated Preview 3 release which will include additional features required for training some of the other benchmarks discussed in the paper. We expect to release the Isaac Gym Preview 3 release by early October, and we will simultaneously update the IsaacGymEnvs github page to include the following benchmarks:
>
> - Ant
> - Humanoid
> - Ingenuity
> - ShadowHand (basic)
> - ShadowHand (DR / Asymmetric OpenAI observations with both FF and LSTM networks)
> - Anymal (both flat and rough terrain)
> - Allegro Hand (basic)
> - Trifinger
>
> We expect to be able to share the AMP and Franka Cube Stacking environments shortly after this initial environment push. All of the benchmark environment code will be available under a BSD license.

---

### Official Review · Reviewer_tjX1 · 2021-09-21
**High-performance training platform for reinforcement learning tasks**

**Rating:** 7
**Confidence:** 2
**Correctness:** Yes
**Clarity:** The paper is well written and easy to…

**Strengths:**

Providing faster, more efficient implementation would lower barrier for RL simulation tasks, and achieve new results.

Thorough evaluation on various RL tasks and simulators.

An API to interface with existing work in this domain.

**Weaknesses:**

Comparison to CPU clusters is not quantified: can authors compare costs of A100 GPU vs a CPU cluster, or perhaps compare FLOPs?

GPU accelerations introduced in Isaac Gym might not provide perf improvements for simpler tasks and fewer environments.

**Additional Feedback:**

L120: “changing the number of environments from 256 to 8192 -- an increase by 5 orders of magnitude --” must by a typo here?

**Documentation:**

The level of detail is sufficient.

**Ethics:**

No ethics concerns

**Relation To Prior Work:**

The comparison to prior work is clear.

**Summary And Contributions:**

The paper introduces a high-performance training platform for a wide variety of reinforcement learning tasks. The key innovation is to run training end-to-end on GPU, reducing CPU-bound bottlenecks (for instance, when a simulation was on GPU but physics state is on CPU), achieving 2-3 orders of magnitude speedup in training times. Isaac Gym provides an optimized implementation which vectorizes observations and actions on GPU allowing to take advantage of the parallelization provided by the simulator.

The speedups are measured and reported on a wide variety of simulation RL tasks, including Ant, Humanoid, and Shadow Hand. The experiments are run on a single A100 GPU.

---

> ### Author Response · Authors · 2021-09-30
> **Reponse to Reviewer tjX1**
>
> Thank you for your positive review. We appreciate your questions and answer them below:
>
> **Comparison to CPU clusters is not quantified: can authors compare costs of A100 GPU vs a CPU cluster, or perhaps compare FLOPs?**
>
> In comparison to a large number of CPU cores, GPUs are not expensive on cloud computing resources. As an example, on Google Compute (https://cloud.google.com/compute/all-pricing), a 128-core CPU costs `$4.32/hour` while a machine with an NVIDA A100 GPU costs `$3.67/hour`. It should be noted that in our results a single GPU was used for both simulation and training. In a CPU based simulation environment GPUs are still usually needed for training neural networks, particularly with large batch sizes.
>
> If we consider the environment used by OpenAI for training the Shadow Hand dexterous manipulation task, they used 384 systems with 16 CPU cores each, in addition to 8 NVIDIA GPUs to reach 37 consecutive successes taking nearly 17 hours. Costs on Google Compute for 17 hours of training time in this setup (enough to train their LSTM model) would be:
>
> CPU portion `$3,499.60` (384 e2-standard-16 systems x 17 hours)
>
> GPU portion: `$499.58` (1 a2-highgpu-8g system x 17 hours)
>
> Total: `$3999.18`
>
> By comparison, 6 hours of A100 compute time on Google Compute (enough to train the Shadow Hand OpenAI LSTM model with Isaac Gym) would cost `$22.04` - a roughly 180x cost reduction.
>
> As noted at line 354, we also estimate that training the OpenAI ShadowHand task on GPU with Isaac Gym would consume about 1/300 the energy required for the equivalent CPU cluster.
>
> **GPU accelerations introduced in Isaac Gym might not provide perf improvements for simpler tasks and fewer environments.**
>
> For the maximum possible acceleration the whole GPU should be utilized - depending on the environment complexity that can require running more than 10K environments in parallel. Simpler tasks may not require as many environments to train to good results, so acceleration rates might be lower, but still about an order of magnitude. In our experiments GPUs were faster than CPUs starting from around 8-16 environments.

---

### Decision · Program_Chairs · 2021-10-10

**Decision:**

Accept

**Comment:**

This paper proposes a very performant physics engine for robotics implemented on the GPU. The main benefit of this approach is that it enables end-to-end GPU training, which reduces the CPU to GPU communication and transfers. The speedups are measured on a wide range of RL tasks.

The paper is well-written and the overall feedback from the reviewers were positive. However, reviewers raised some concerns in their reviews. I recommend the authors to take those comments into account and try to address them in the camera-ready version of this paper.